# Primary Clear Cell Carcinoma of the Pancreas: A Rare Case Report

**DOI:** 10.3390/diagnostics12092046

**Published:** 2022-08-24

**Authors:** Yoo-Na Kang

**Affiliations:** Department of Forensic Medicine, School of Medicine, Kyungpook National University, 680 Gukchaebosang-ro, Jung-gu, Daegu 41944, Korea; yoonakang@knu.ac.kr

**Keywords:** pancreas, primary, clear cell carcinoma, hepatitis, hepatocellular carcinoma

## Abstract

Most pancreatic carcinoma is ductal adenocarcinoma. Primary pancreatic clear cell carcinomas composed almost entirely of clear tumor cells are very rare. We present a case of a 72-year-old man with a pancreatic mass, which was detected on abdominal computed tomography (CT). He had no symptoms and no abnormal findings on physical examination; however, he had a history of hepatitis B, hepatitis C, and hepatocellular carcinoma. He had received anti-viral treatment and radiofrequency ablation twice until 2 years prior. One year prior, follow-up contrast-enhanced abdominal CT revealed a newly developed pancreatic mass. Laparoscopic radical antegrade modular pancreato-splenectomy was performed. An ill-defined white-to-tan firm solid mass was observed in the pancreas, approximately 4.3 cm in diameter. The tumor cells showed >95% clear cell features, with a large round to oval nuclei and abundant clear cytoplasms, and well-defined cell membranes. Immunohistochemical staining revealed that the tumor cells were positive for cytokeratin 7, cytokeratin 19, HNF-1β, MUC-1, and p53. We excluded the possibility of metastatic clear renal cell carcinoma, neuroendocrine carcinoma, perivascular epithelioid cell tumor, malignant melanoma, and sarcoma because of the negativity for vimentin, chromogranin, synaptophysin, and HMB45. Consequently, he was diagnosed as having primary clear cell carcinoma of the pancreas and was treated with postoperative radiotherapy. Two months later, abdominal CT was suspicious for local recurrence at the resection margin. Additional adjuvant FOLFIRINOX chemotherapy was carried out 12 times. The patient is still alive after his third radiofrequency ablation for the newly-developed hepatic mass. Immunohistochemical staining for MUC-1 and HNF-1β, as well as histologic feature is very helpful for the diagnosis of primary pancreatic clear cell carcinoma with imaging methods for metastasis exclusion.

**Figure 1 diagnostics-12-02046-f001:**
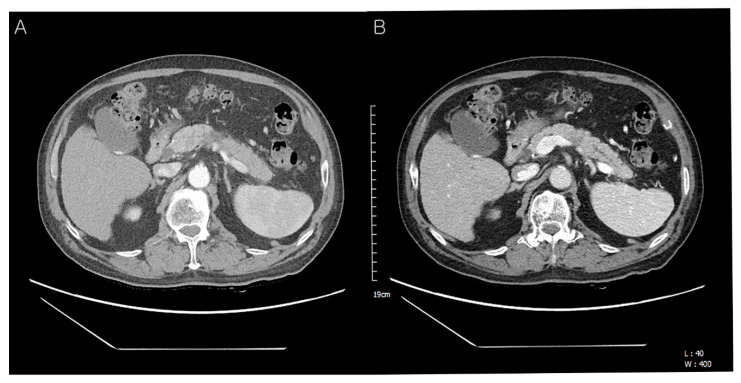
Preoperative abdominal computed tomography (CT) findings. A 72-year-old man presented with a pancreatic mass on abdominal CT. However, he had no other symptoms of jaundice, abdominal pain, or weight loss. On physical examination, the liver and spleen were not enlarged and the vital signs were stable. He was diagnosed with hepatitis B and C 30 and 10 years ago and had received anti-viral treatment (Table 1). Subsequently, he was diagnosed with hepatocellular carcinoma in segment 7 and treated with radiofrequency ablation (RFA). Eight years later, an additional RFA treatment was performed for the newly developed hepatic mass in segment 8 (Table 1). The laboratory data were as follows: total protein 6.7 g/dL (normal: 6.6–8.7 g/dL), albumin 4.0 g/dL (normal: 3.5–5.2 g/dL), total bilirubin 0.37 mg/dL (normal: <1.2 mg/dL), direct bilirubin 0.2 mg/dL (normal: 0.01–0.3 mg/dL), serum AST 17 U/L (normal: ≤40 U/L), ALT 19 U/L (normal: ≤41 U/L), serum CEA 2.3 ng/mL (normal: 0–7.0 ng/mL), serum CA19-9 121.0 U/mL (normal: 0–37.0 U/mL), amylase 98 U/L (normal: 28–110 U/L), lipase 91 U/L (13–60 U/L), BUN 16.3 mg/dL (normal: 6–20 mg/dL), creatinine 0.58 mg/dL (normal: 0.7–1.2 mg/dL). Follow-up contrast-enhanced abdominal CT showed an approx. 4.0 cm-sized hypodense mass in the distal body and tail of the pancreas with an irregular margin (**A**). The upstream pancreatic duct dilatation was also detected (**B**). Hence, the patient’s condition was diagnosed as pancreatic ductal adenocarcinoma preoperatively.

**Table 1 diagnostics-12-02046-t001:** Clinical History of Patient.

Date	History
1992	Chronic hepatitis B diagnosis, anti-viral treatment
2011	Chronic hepatitis C diagnosis, anti-viral treatment
2011.12.	CT, MRI: 2.5cm hepatic mass(S7)-hepatocellular carcinoma diagnosis
2011.12.	Radiofrequency ablation (RFA) #1
2019.12.	CT, MRI: 2.4cm hepatic mass(S8)-hepatocellular carcinoma diagnosis
2019.12.	Radiofrequency ablation (RFA) #2
2021.3.	CT, MRI: 1.5cm low-density pancreatic mass-pancreatic cancer diagnosis
2021.4.	Laparoscopic radical antegrade modular pancreato-splenectomy
2021.6.	CT: Suspicious of local recurrence at the resection margin of pancreas
2021.7.–2022.2.	Post-operative radiotherapy and FOLFIRINOX chemotherapy #1–#12
2022.3.	CT: A new-developed hepatic mass(S8)-hepatocellular carcinoma diagnosis
2022.3.	Radiofrequency ablation (RFA) #3

S: segment, MRI: magnetic resonance imaging, CT: computerized tomography.

**Figure 2 diagnostics-12-02046-f002:**
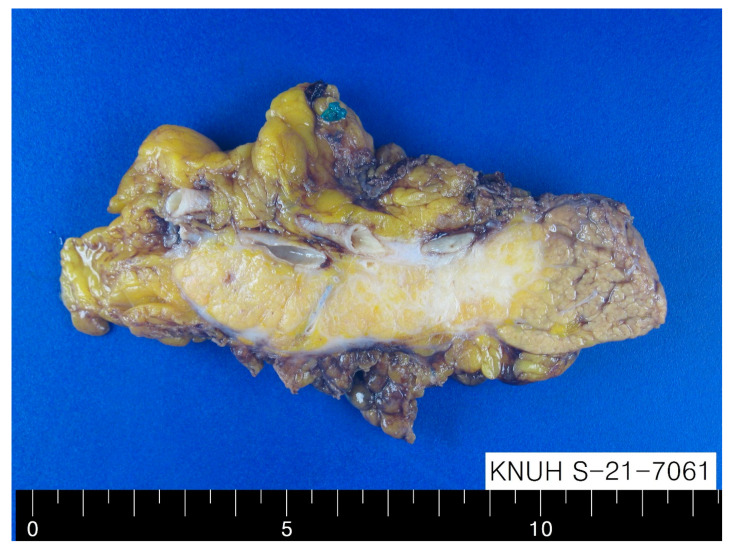
Gross finding of dissected pancreatic mass. Radical antegrade modular pancreato-splenectomy was performed. Gross findings showed an ill-defined pale tan, firm, and solid mass without areas of necrosis and hemorrhage in the pancreatic body, measuring 4.3 cm in maximum diameter (Figure 2).

**Figure 3 diagnostics-12-02046-f003:**
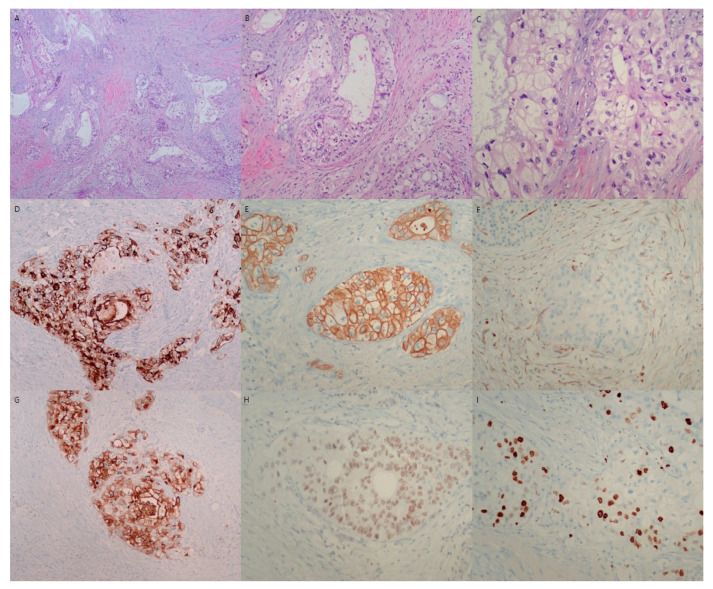
Microscopic findings of resected pancreatic mass. Microscopic evaluation of the pancreatic mass shows variable tubular growth of tumor cells in a desmoplastic background (H&E, ×40) (**A**). The variable tubules are all composed of clear tumor cells (H&E, ×100) (**B**). The characteristic clear tumor cells have large round to oval pleomorphic nuclei and abundant clear cytoplasm (H&E, ×200) (**C**). Microscopically, clear cell carcinoma of the pancreatic mass is composed of large round to oval cells with hyperchromatic nuclei and abundant clear cytoplasm with distinct cell margins. The tumor cells showed tubules, cords, solid trabeculae, and solid growth patterns. Hobnail patterns of clear tumor cells were also occasionally observed. The nuclei of the tumor cells were small, irregular, pleomorphic, and often eccentrically placed. In this case, clear cells were present in most areas of the tumor mass (>95%). Immunohistochemically, the tumor cells were positive for cytokeratin 7 (**D**), cytokeratin 19 (**E**), MUC-1 (**G**), HNF-1β (**H**), and Ki-67 (**I**), and negative for vimentin (**F**), chromogranin, synaptophysin, and CD56. Consequently, the patient was diagnosed with primary pancreatic clear cell carcinoma with tumor invasion to peripancreatic fat tissue, lympho-vascular invasion and multiple peripancreatic nodal metastasis (T3N2M0, stage III). The resection margin of the pancreas was free of tumor.

**Figure 4 diagnostics-12-02046-f004:**
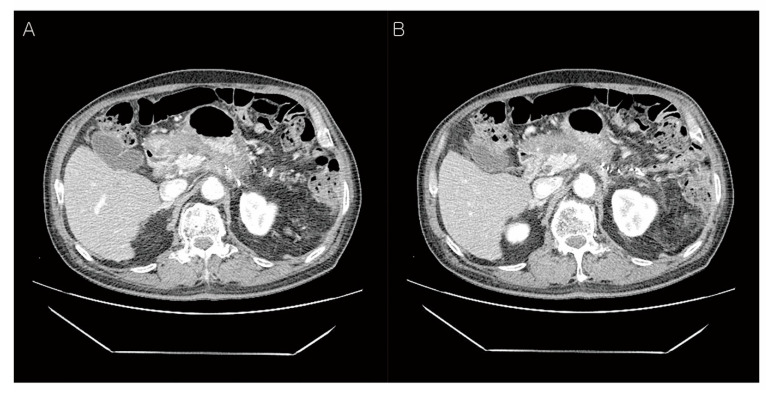
Abdominal CT findings post-operation, two months later. The patient underwent postoperative radiotherapy for three weeks. After two months, abdominal CT findings showed more prominent soft tissue thickening adjacent to the common hepatic artery (**A**) and 3.8 cm loculated fluid collection at the resection margin of the pancreas (**B**). This CT finding showed local recurrence at the resection margin of the pancreas, therefore, adjuvant FOLFIRINOX chemotherapy was Carried out 12 times for nine months.

**Figure 5 diagnostics-12-02046-f005:**
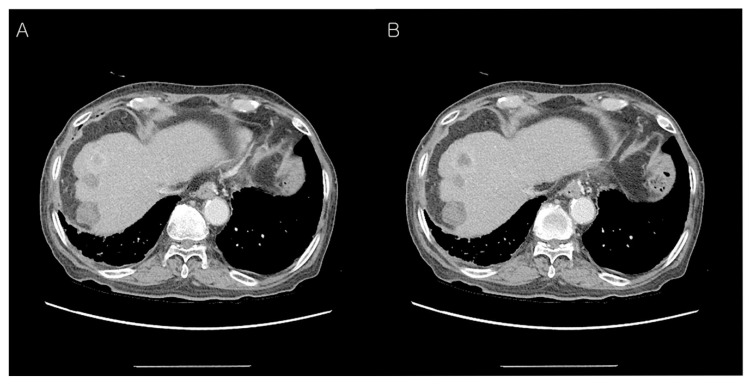
Abdominal CT findings post-operation, 10 months later. Ten months after the operation, the follow-up CT revealed a heterogenous enhancing mass (upper) in segment 8 of the liver with two previous RFA-treated hepatic masses (middle and lower) (**A**,**B**). After the third radiofrequency ablation for the new-developed hepatic mass, the patient is still alive.

Most pancreatic carcinoma is ductal adenocarcinoma. Cubilla et al. first described pancreatic clear cell carcinoma in 1980 [1]; subsequently, only a few cases of pancreatic clear cell carcinoma have been reported [2,3,4,5,6,7]. Currently, the World Health Organization (WHO) recognizes clear cell carcinoma as a “miscellaneous” carcinoma of the pancreas [8]. Among pancreatic adenocarcinoma, 24% is identified as mixed ductal adenocarcinoma with focal clear cell features [9], however, primary pancreatic clear cell carcinoma with more than 95% clear cell component is very rare [9]. Therefore, this case of primary clear cell carcinoma is meaningful to define the characteristics and prognosis of this very rare tumor.

Clear cell carcinoma mostly has characteristic clear tumor cells. Differential identification from conventional ductal adenocarcinoma is needed due to the difference in the treatment and prognosis as well as morphologic characteristics. Sasaki et al. described clear tumor cells as those with abundant clear cytoplasms and centrally or peripherally located round nuclei without prominent nucleoli [5]. Ray et al. described that pancreatic clear cell carcinoma had abundant clear cells arranged in nests, cords, or tubules with clear cytoplasms, distinct cell borders, and eccentric nuclei [10]. In this case of pancreatic clear cell carcinoma, hematoxylin and eosin (H&E) staining revealed variable-sized nests and sheets as well as glands composed of large clear cells. The tumor cells had atypical hyperchromatic nuclei and abundant clear cytoplasms (Figure 3A–C). According to a previous study, the vacuoles of clear tumor cells were devoid of glycogen and mucin [9]. PAS, D-PAS, and mucicarmine staining confirmed that the clear cytoplasm was not due to the accumulation of glycogen or mucin [9].

For the differential diagnosis of primary pancreatic clear cell carcinoma, metastatic carcinoma, neuroendocrine carcinoma, perivascular epithelioid cell tumor (sugar tumor), malignant melanoma, and sarcoma should be considered. Metastatic renal clear cell carcinoma (RCCC) can be found in the pancreas. RCCC is usually composed of clear cell nests with scanty fibrovascular stroma. Although metastasis of RCCC is more common in the lung and liver, it may also rarely be found in other metastatic sites, such as the duodenum, kidneys, adrenal glands, and pancreas. Recently, a few cases of renal clear cell carcinoma was reported with metastasis to the pancreas [11]. Metastasis in the pancreatic gland is infrequent, representing between 2–5% of the tumors that affect this organ [12]. To date, only a few cases of breast carcinoma, colon carcinoma, malignant melanoma, and sarcoma have been reported [13]. To confirm the diagnosis of metastatic carcinoma, the presence of other extra-pancreatic primary tumors is essential. In this case, clinical and radiological investigations by ultrasound sonography, magnetic resonance imaging (MRI) and positron emission tomography-computed tomography (PET-CT) could not detect any extra-pancreatic primary tumors. Some endocrine neoplasms of the pancreas display a predominantly solid growth pattern of monotonous round tumor cells with clear cell features. In this case, the possibility of pancreatic endocrine tumors was excluded because of the absence of neuroendocrine markers, such as chromogranin and synaptophysin in tumor cells. Perivascular epithelioid cell tumors (sugar tumors) show positivity for HMB-45 on immunohistochemical staining and rarely occur in the pancreas. However, it could be excluded the possibility of perivascular epithelioid cell tumors and malignant melanoma with sarcomas because of the negativity of immunohistochemical staining for HMB-45 and vimentin.

Pancreatic carcinoma has a high mortality rate, as the pancreatic malignancy generally remains asymptomatic until it reaches an advanced stage. Therefore, the exact diagnosis, as well as early detection, would be necessary. Kim et al., Lüttges et al., and Ray et al. classified clear cell carcinoma as tumors with more than 75%, 90%, and 95%, respectively, of the tumor cells being clear cells [4,9,10]. The diagnostic criteria, that is, the amount of clear tumor cell components was different in the previous reports. 

Certain criteria to define clear cell carcinoma of the pancreas have not yet been established [2]. MUC-1 as well as cytokeratins (CK) 7,8,18,19 are overexpressed in pancreatic ductal carcinoma, with a predominantly membranous and variable cytoplasmic staining pattern [4,14,15]. In this case, the clear tumor cells showed positivity of MUC-1 (Figure 3G), cytokeratin 7 (Figure 3D), and cytokeratin 19 (Figure 3E), and showed negativity of vimentin (Figure 3F) and neuroendocrine markers, which indicate pancreatic clear cell carcinoma with ductal phenotype. K-ras analysis revealed showed contradictory results. K-ras analysis showed a point mutation at codon 12 in clear cell carcinoma of pancreas [4,6], while no mutation at codon 12 in either clear cell or duct-like components [5]. DNA microarray analysis revealed a transcription profile clearly differing from that of normal pancreatic tissue and pancreatic ductal adenocarcinoma [16]. Molecularly, it is still unclear whether clear cell carcinoma is a subtype of ductal adenocarcinoma. Due to the small number of cases, more studies associated with treatment and prognosis are needed in the future. All clear cell carcinomas of the pancreas showed strong staining of hepatocyte nuclear factor 1-beta (HNF-1β), whereas the majority of conventional ductal adenocarcinoma in the pancreas showed no or weak HNF-1β positivity [9]. High expression of HNF-1β showed statistically significant worse survival (*p* < 0.01) [9]. In this case, the clear tumor cells showed nuclear positivity of HNF-1β (Figure 3H), and post-operation, two months later, local recurrence at the resection margin of the pancreas occurred (Figure 4). Although the immunohistochemical staining of HNF-1β as the diagnostically useful marker is not sufficient due to the lack of cases, the positivity of MUC-1 and HNF-1β, as well as histologic feature, is very helpful for the diagnosis of primary pancreatic clear cell carcinoma with imaging methods for metastasis exclusion.

## Data Availability

Not applicable.

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
