# Peer review of "Primary Clear Cell Carcinoma of the Pancreas: A Rare Case Report"

_diagnostics, 2022, doi:10.3390/diagnostics12092046_

Round 1

Reviewer 1 Report

This is a case report which describes a case of primary clear cell carcinoma of the pancreas. I recommend several major revisions.

1. In the abstract, you stated that the patient had received Zeffix. Please avoid using brand name.

2. In the abstract and main text, you stated that the patient received a folfirinox regimen. Chemotherapy regimens should be written in capital letters (eg, FOLFIRINOX).

3. There is a serious discrepancy regarding the postoperative recurrence. You described that hepatic metastasis developed 11 months after surgery in the abstract. However, in the main text, you stated that local recurrence in the pancreas developed 10 months after surgery. What is correct?

4. Please provide relevant images regarding the postoperative recurrence.

5. The radiological part should be improved. You stated that abdominal CT showed an irregular hypodense mass in the distal body and tail of the pancreas. Please show this CT image with figure legend about details of lesion (e.g. mass size, involvement of vessels).

6. Did you stage the tumor by CT scan only? What modality did you use to rule out distant metastasis? Please show relevant images for staging the tumor.

7. Did you perform EUS-FNA preoperatively for the pathologic diagnosis?

8. The past history, family history, and laboratory findings were too long. Please remove normal findings and focused on abnormal findings.

9. Please provide author contributions, funding, IRB statement, and informed consent… following the journal instruction.

Author Response

Thank you for your mentions. 

My answers: 

  1. In the abstract, you stated that the patient had received Zeffix. Please avoid using brand name.
  2. A) I deleted Zeffix …
  3. In the abstract and main text, you stated that the patient received a folfirinox regimen. Chemotherapy regimens should be written in capital letters (eg, FOLFIRINOX).
  4. A) I changed folfirinox to FOLFIRINOX
  5. There is a serious discrepancy regarding the postoperative recurrence. You described that hepatic metastasis developed 11 months after surgery in the abstract. However, in the main text, you stated that local recurrence in the pancreas developed 10 months after surgery. What is correct?
  6. A) I corrected these discrepancy.

Local recurrence in the pancreas developed 2 months after surgery.

Hepatic new-developed mass was detected 10 months after surgery.

  1. Please provide relevant images regarding the postoperative recurrence.
  2. A) I added more images suspicious of postoperative recurrence.
  3. The radiological part should be improved. You stated that abdominal CT showed an irregular hypodense mass in the distal body and tail of the pancreas. Please show this CT image with figure legend about details of lesion (e.g. mass size, involvement of vessels).
  4. A) I added abdominal CT scan showing an irregular hypodense mass in the body of the pancreas.
  5. Did you stage the tumor by CT scan only? What modality did you use to rule out distant metastasis? Please show relevant images for staging the tumor.
  6. A) In the course of initial work-up and follow-up, MRI and PET CT as well as Abdominal CT were done many times. But the images with positive finding were dealed in the paper,
  7. Did you perform EUS-FNA preoperatively for the pathologic diagnosis?
  8. A) EUS-FNA did not performed preoperatively.
  9. The past history, family history, and laboratory findings were too long. Please remove normal findings and focused on abnormal findings.
  10. A) I removed partial portions of normal findings.
  11. Please provide author contributions, funding, IRB statement, and informed consent… following the journal instruction.
  12. A) I added funding, IRB statement, and informed consent.

Reviewer 2 Report

The manuscript is of interest, nonetheless,

1.I would suggest to restructure as follows:

Part 1 — Working Title, WHAT happened: Timeline and Narrative

Develop a descriptive and succinct working title that describes the phenomenon of greatest interest (symptom, diagnostic test, diagnosis, intervention, outcome).

WHAT happened. Gather the clinical information associated with patient visits in this this case report to create a timeline as a figure or table. The timeline is  a chronological summary of the visits that make up the episodes of care from this case report.

Narrative of the episode of care (including tables and figures as needed).

The presenting concerns (chief complaints) and relevant demographic information.

Clinical findings: describe the relevant past medical history, pertinent co-morbidities, and important physical examination (PE) findings.

Diagnostic assessments: discuss diagnostic testing and results, a differential diagnosis, and the diagnosis.

Therapeutic interventions: describe the types of intervention (pharmacologic, surgical, preventive, lifestyle) and how the interventions were administered (dosage, strength, duration, and frequency). Tables or figures may be useful.

Follow-up and outcomes: describe the clinical course of the episode of care during follow-up visits including (1) intervention modification, interruption, or discontinuation; (2) intervention adherence and how this was assessed; and (3) adverse effects or unanticipated events. Regular patient report outcome measurement surveys such as PROMIS® may be helpful.

Part 2 — WHY it might have happened: Introduction, Discussion, Conclusion

The introduction should briefly summarize why this case report is important and cite the most recent CARE article (Riley DS, Barber MS, Kienle GS, AronsonJK,  et al. CARE guidelines for case reports: explanation and elaboration document. JClinEpi 2017 Sep;89:218-235. doi: 10.1016/jclinepi.2017.04.026).

WHY it might have happened. The discussion describes case management, including strengths and limitations with scientific references.

The conclusion, usually one paragraph, offers the most important findings from the case without references.

Part 3 — Abstract, Keywords, References, Acknowledgements, and Informed Consent

Abstract. Briefly summarize in a structured or unstructured format the relevant information without citations. Do this after writing the case report. Information should include: (1) Background, (2) Key points from the case; and (3) Main lessons to be learned from this case report.

Keywords. Provide 2 to 5 keywords that will identify important topics covered by this case report.

References. Include appropriately chosen references from the peer-reviewed scientific literature.

Acknowledgements. A short acknowledgements section should mention funding support or conflicts of interest, if applicable.

Informed Consent and Patient Perspective. The patient should provide informed consent (including a patient perspective) and the author should provide this information if requested. Some journals have consent forms which must be used regardless of informed consents you have obtained. Rarely, additional approval (e.g., IRB or ethics commission) may be needed. The patient should share their perspective on the treatment(s) they received in one to two paragraphs. It is often best to ask for informed consent and the patient’s perspective before you begin writing your case report.

Appendices (If indicated).

2. This reviewer personally misses some insights regarding pathobiological mechanism of PDAC malignant evolution. Genetic alterations, especially the K-Ras mutation, carry the heaviest burden in the progression of pancreatic precursor lesions into pancreatic ductal adenocarcinoma (PDAC). The tumor microenvironment is one of the challenges that hinder the therapeutic approaches from functioning sufficiently and leads to the immune evasion of pancreatic malignant cells. Mastering the mechanisms of these two hallmarks of PDAC can help us in dealing with the obstacles in the way of treatment (please refer to PMID: 33918146 and expand):

The underlying message here is that more precision and individualized

approaches need to be tested in well designed clinical trials – a

challenge, but I would be interested in their perspective of how this

might be done.

Author Response

Thank you for your mentions. 

My answers: 

1.

Part 1 — Working Title, WHAT happened: Timeline and Narrative

A) I corrected the title because this case is very rare primary pure clear cell carcinoma of pancreas.   Also I made the timeline of history as a table.

I described the clinical course of symptom, laboratory, radiologic, diagnostic, and therapeutic approach.   

Part 2 — WHY it might have happened: Introduction, Discussion, Conclusion

A) I made a conclusion:

Early detection and exact diagnosis of variable pancreatic cancers would be necessary.  The positivity of MUC-1 and HNF-1β of clear tumor cells with clinical evaluation for the exclusion of metastasis is very helpful for the diagnosis.

The additional information of more primary pancreatic pure clear cell carcinomas (such as prognosis and gene analysis as well as diagnostic tool)

would be necessary to define this disease entity.

Part 3 — Abstract, Keywords, References, Acknowledgements, and Informed Consent

A) I summarize the abstract briefly, and I choose a few new key words associated with the history of the patient. 

Appendices (If indicated).

A) Clear cell carcinoma of pancreas with focal clear cell change is rarely seen. Among them, pure clear cell carcinoma is very rare. Recently clear cell carcinoma classification was made and the research of diagnosis and prognosis as well as treatment is processing. So I think more additional information of primary pancreatic pure clear cell carcinomas would be necessary to define this disease entity.

Round 2

Reviewer 1 Report

I appreciate the revisions performed. The paper much improved but several minor revisions were still needed.

In Line 24, please state clearly that the local recurrence has occurred in the pancreas.

In figures 1,4,5, the right side of the CT scan was cut off, please adjust.

Author Response

In Line 24, please state clearly that the local recurrence has occurred in the pancreas.

Answer )  I changed the sentence ….  This findings showed local recurrence at the resection margin of pancreas.

In figures 1,4,5, the right side of the CT scan was cut off, please adjust.

Answer ) I made figure 1,4,5 again without trimming the margins.  These are the original photos.(attached file) 

Reviewer 2 Report

The author has clarified several questions I raised in my previous review. Unfortunately, most of the major problems have not been addressed by this revision: no marked-up version with the performed changes is provided (it should), and several of the suggestions made to improve the manuscript quality (especially the discussion criticism) have been neglected (see previous comments).

Author Response

The author has clarified several questions I raised in my previous review. Unfortunately, most of the major problems have not been addressed by this revision: no marked-up version with the performed changes is provided (it should), and several of the suggestions made to improve the manuscript quality (especially the discussion criticism) have been neglected (see previous comments).

Answer )

I am sorry not to attach marked-up version.

So, I reviewed it again more in detail.

  • Marked-up version with the performed changes is attached

: I corrected the title because this case is very rare primary pure clear cell carcinoma of pancreas.

 I made the timeline of history as a table.

I described the clinical course of symptom, laboratory, radiologic, diagnostic, and therapeutic approach.

I described each sections separately … that is Introduction, Discussion, Conclusion

I summarize the abstract briefly, and I choose a few new key words associated with the history of the patient.

        I added chief complaint and the result of physical examination

        I described therapeutic interventions of this case. But there is enough data associated with prognosis and survival, due to rare clear cell carcinoma.

        I added follow-up CT and additional therapeutic methods with patient’s status.

  1. The conclusion, usually one paragraph, offers the most important findings from the case without references.

Answer) I made a conclusion:

Positivity of MUC-1 and HNF-1β as well as histologic feature with the exclusion of metastasis is very helpful for the diagnosis of primary pancreatic clear cell carcinoma

Molecularly, it is still unclear whether clear cell carcinoma is a subtype of ductal adenocarcinoma. Due to the small number of cases, more studies are needed in the future.

  1. I revised abstract including background, key points from the case and main lessons to be learned from this case report.

Background : Most of pancreatic carcinoma is ductal adenocarcinoma. Primary pancreatic clear cell carcinomas composed almost entirely of clear tumor cells are very rare.

Key point: The tumor cells showed > 95 % clear cell features, with large round to oval nuclei and abundant clear cytoplasms, and well-defined cell membranes. Immunohistochemical staining revealed that the tumor cells were positive for cytokeratin 7, cytokeratin 19, HNF-1β, MUC-1, and p53.

Lesson: Immunohistochemical staining for MUC-1 and HNF-1β as well as histologic feature is very helpful for the diagnosis of primary pancreatic clear cell carcinoma with imaging methods for metastasis exclusion.

  1. I provided 5 keywords, important topics covered by this case report.
  2. I chosed main related references.
  3. I added informed Consent and Patient Perspective.
  4. I added the portion of genetic alterations, especially the K-Ras mutation.

Round 3

Reviewer 2 Report

I still do not see any marked-up MODIFIED version and can not detect any rebuttal or point by point response to my previous revision (this makes the manuscript subpar). For the author's convenience I recall: The author has clarified several questions I raised in my previous review. Unfortunately, most of the major problems have not been addressed by this revision: no marked-up version with the performed changes is provided (it should), and several of the suggestions made to improve the manuscript quality (especially the discussion criticism) have been neglected (see previous comments).